# A Fluorescent Nanosensor for Silver (Ag^+^) and Mercury (Hg^2+^) Ions Using Eu (III)-Doped Carbon Dots

**DOI:** 10.3390/nano12030385

**Published:** 2022-01-25

**Authors:** Cátia Correia, José Martinho, Ermelinda Maçôas

**Affiliations:** Centro de Química Estrutural (CQE) and Institute of Molecular Science, Instituto Superior Técnico, Universidade de Lisboa, 1049-001 Lisboa, Portugal; catia.correia@tecnico.ulisboa.pt (C.C.); jgmartinho@tecnico.ulisboa.pt (J.M.)

**Keywords:** sensor, fluorescence, ratiometric, silver, mercury, carbon dots, europium

## Abstract

Carbon dots doped with Eu^3+^ ions (Eu-Cdots) were prepared by a hydrothermal treatment, using citric acid and urea as precursors and Eu (NO_3_)_3_ as a europium source. The Eu^3+^ ions are strongly coordinated with the carboxylate groups at the surface of the Cdots and incorporated within the nanographene network in the carbon core. Vibrational spectroscopy provides evidence of such interaction with identification of bands assigned to the stretching of the Eu-O bond. Eu^3+^ doped Cdots have larger diameters then undoped Cdots, but they are divided into smaller domains of sp^2^ carbon. The UV-vis excitation spectrum provides evidence of energy transfer from the Cdots to the Eu^3+^. The luminescence spectrum shows the characteristic sharp peaks of Eu^3+^ in the red part of the visible spectrum and a broad emission of Cdots centered at 450 nm. The luminescence of the Cdots is strongly quenched by Hg^2+^ and Ag^+^, but not by other cations. The quenching mechanism differs significantly depending on the nature of the ion. Both the blue emission of Cdots and the red emission of Eu^3+^ are quenched in the presence of Hg^2+^ while only the emission of the Cdots is affected by the presence of Ag^+^. A ratiometric sensor can be built using the ratio of luminescence intensities of the Cdots to the Eu^3+^ peaks.

## 1. Introduction

Carbon dots (Cdots) are a new class of carbon nanomaterials with interesting optical properties, such as high emission yields, excitation wavelength dependent emission, and high photostability. When combined with secondary properties, such as low toxicity, chemical stability, and water solubility, this makes them uniquely suited for applications in bioimaging, sensing, nanomedicine, catalysis, lightning, and photonics [1,2,3,4,5,6]. Cdots can be easily prepared by top–down and bottom–up approaches using low-cost and environmentally-friendly materials. Moreover, their physicochemical properties can be controlled and tuned by doping, surface passivation and functionalization. The photoluminescence of Cdots can be employed to build optical sensors for metal cations (e.g., Fe^3+^, Cu^2+^, Ag^+^, Hg^2+^, Pb^2+^), anions (e.g., NO^2−^, S^2−^, Cl^−^, P_2_O_7_^4−^), and molecular species (e.g., amino acids, vitamins, carbohydrates) [7]. Generally, the carbon dots need to be doped or appropriately functionalized to probe selectively the specific analyte. In many cases the analyte induces the quenching of the fluorescence (turn-off), while in others the fluorescence can be enhanced (turn-on) by suppressing the quenching or by generating new fluorescent species. A typical example of a turn-off-on mechanism is that of the detection of pyrophosphates (P_2_O_7_^4−^) by nitrogen-doped carbon dots in the presence of Eu^3+^. The Eu^3+^ ions initially linked to the surface groups of Cdots suppress the luminescence, which is restored in the presence of P_2_O_7_^4−^ due to the higher affinity of Eu^3+^ to the pyrophosphates than to the carboxylic and amido groups onto the surface of the Cdots [8].

Silver is a very toxic heavy metal generated in diverse activities like electronics, photography, and pharmaceutics. To meet the demand for simple and fast determination of Ag^+^ in aquatic environments and organisms, both turn-on and turn-off Cdots based sensors have been proposed [9,10,11,12,13,14,15,16,17,18]. The sensing mechanism remains quite elusive, with the fluorescence quenching by Ag^+^ being explained by complexation via the carboxylic [11], and the oxygen/nitrogen containing surface groups of Cdots [14], or the enhanced interactions with S; S, N, and N, F-doped Cdots [12,13,18], and also the coordination reaction between C=N surface groups of formamide-bases of N-doped Cdots and Ag^+^ [16,17]. The fluorescence enhancement of the Cdots in the presence of Ag^+^ has been attributed to the formation of silver clusters and/or complexation of silver ions by amines at the surface of the Cdots [9,10]. The amine and/or phenol hydroxyl groups on the surface of Cdots were shown to be able to reduce Ag^+^ to elemental Ag (0) with the formation of silver nanoparticles detected by resonance light scattering [19]. 

Only a few Ag^+^ ratiometer sensors were proposed until now [17,20,21]. One of them uses, as a sensor, formamide-derived N-doped Cdots encapsulated in the cavities of Eu (III)-based metal-organic frameworks (Eu-MOFs). The Cdots luminescence is quenched by Ag^+^, while the Eu^3+^ emission remains almost constant, working as an internal standard [17]. An alternative approach is the use of a mixture of Cdots and o-phenylenediamine [20]. The addition of Ag^+^ to the mixture oxidizes the non-fluorescent o-phenylenediamine to the fluorescent 2,3-diaminophenazine by the reduction of Ag^+^ to Ag (0). The emission of Cdots is then quenched by energy transfer to diaminophenazine resulting in the increase of the fluorescence intensity ratio of diaminophenazine to Cdots. In addition, a dual emitting nanohybrid composed of Si,N-co-doped carbon dots and N-acetyl-L-cysteine-capped CdTe quantum dots was proposed as an Ag^+^ ratiometric sensor based on the quenching of the quantum dots luminescence by Ag^+^ and using as internal standard the Cdots fluorescence that remain almost constant [21].

The mercury cation (Hg^2+^) can be released into the environment through a variety of anthropogenic and natural sources. The Hg^2+^ is a potent neurotoxin that can cause severe damage to the central nervous system and interfere with mitochondrial activity [22,23]. Several analytical techniques have been used to detect Hg^2+^, those based on Cdots fluorescence quenching being very appealing due to their rapid response, high sensitivity, and low cost [24,25,26,27,28]. To increase selectivity, the Cdots surface has been engineered to inhibit the interference of other metal ions that bind to surface functional groups and thymine-rich DNA was covalently linked to the surface to specifically recognize the mercury ions [27]. An alternative strategy used a Tb^3+^-Cdots/3-aminophenylboronic acid hybrid with dual emission from the Cdots and the Tb^3+^ ion to detect simultaneously NO_2_^−^ and Hg^2+^ in environmental water [28]. The 3-aminophenylboronic acid coordinates with Hg^2+^ that quenches selectively the Cdots emission by electron transfer, while the emission of Tb^3+^ remains unchanged. NO_2_^−^ is detected by the quenching of the Tb^3+^ luminescence by electron transfer to NO_2_^−^, while the emission of the Cdots remains unaffected. 

The aim of these work is to develop a fluorescent nanosensor for Ag^+^ and Hg^2+^ ions using Eu^3+^ doped Cdots (Eu-Cdots). The Eu^3+^ ion was chosen among other lanthanide dopants because it is a well-known red light-emitter with high adsorption energy and low diffusion barrier when adsorbed in graphene compared to other lanthanides [29]. The emission of the Eu-Cdots is composed of a broad emission of the Cdots in the blue (450 nm) and the characteristic narrow bands of the red emission of Eu^3+^ ions (>590 nm). The broad emission of the Eu-Cdots is quenched by both cations with a significantly higher efficiency compared to pristine Cdots. The Eu^3+^ emission of the Eu-Cdots is quenched by Hg^2+^ but not by Ag^+^, which can be used to distinguish the two metals. Furthermore, the invariance of the Eu^3+^ emission in the presence of Ag^+^ can be used as an internal standard to build a ratiometer Ag^+^ sensor by measuring the ratio of emission intensities of the Cdots band to the Eu^3+^ peak at non-overlapping wavelengths. This sensor has de advantage of being non-toxic as the quantum dots are considerably more photostable than alternative sensors based on organic fluorophores. 

## 2. Materials and Methods

### 2.1. Materials

All chemicals used were of analytical pure grade. Citric acid (99.5%), urea (99.5%), and Eu (NO_3_)_3_ (99.9%)) were purchased from Sigma Aldrich (Darmstadt, Germany).

### 2.2. Methods

The linear absorption spectra were recorded in a JASCO V-540 spectrophotometer (Cremella, LC, Italy). The fluorescence spectra were recorded using a HORIBA Jobin Yvon Fluorolog 3–22 Spectrofluorometer (Kyoto, Japan) with a 450 W xenon lamp. Typically, the optical properties were measured in solutions of 0.3 g/L of Cdots, 3 g/L of Eu (NO_3_)_3_, 0.8 g/L of Eu-Cdots and a mixture of Cdots plus Eu^3+^ in water containing 3 g/L of Eu (NO_3_)_3_ and 0.03 g/L of Cdots. For the metal titrations, stock solutions of the metals (1–3 mM) were prepared directly in the dispersions of Eu-Cdots, Cdots or Cdots + Eu^3+^ mixture and added systematically directly into quartz cuvettes containing 3 mL of the corresponding dispersions without the metals. This procedure avoids dilution of the dispersion during titration. The fluorescence quantum yields were determined by the reference method using coumarin 153 in ethanol (Φ_F_ = 0.38, λ_em_ = 410–470 nm) [30]. The fluorescence decay curves were recorded at 450 nm by the Single-Photon Timing technique under excitation at 330 nm using a multichannel analyzer working with 1024 channels and time scales of 68.36 ps/channel (Eu-Cdots + Hg^2+^) and 29.30 ps/channel (Cdots, Eu-Cdots and Eu-Cdots + Ag^+^). The excitation source was the second harmonic of a Coherent Radiation Dye laser 700 series (laser dye DCM, 610–680 nm, 130 mW, 5 ps, 4 MHz). The emission was collected at the magic angle relative to the vertical polarized excitation and selected by a Jobin Yvon HR320 monochromator (HORIBA Jobin Yvon, Inc., Kyoto, Japan). The instrument response function for deconvolution (35–80 ps FWHM) was recorded using a scattering dispersion of colloidal silica in water. The FTIR spectra were recorded in KBr pellets (Aldrich, 99%, FT-IR grade) using a Mattson 7000 FT-IR spectrometer (Madison, WI, USA) by accumulating 256 interferograms with a resolution of 8 cm^−1^. Raman characterization was carried out by unpolarized Raman microscopy (HORIBA LabRAM HR 800 Evolution (Kyoto, Japan) using a 532 nm diode laser delivering 10 mW at the sample position and a 600 groves per mm grating. Data were collected in the range of 900–1800 cm^−^^1^ with a resolution of 4 cm^−^^1^ using only 10–20% of the laser power, with acquisition times of 20 s and 20 accumulations using either 50× or 100× objective lenses. Transmission electron microscopy (TEM) images were obtained on a Hitachi transmission electron microscope (Model H-8100 with a LaB6 filament) (Tokyo, Japan) with an acceleration voltage of up to 200 kV. One drop of the dispersion of the Cdots in water was deposited on the TEM grid (ultrathin carbon grid on a lacey carbon support film, 400 mesh (PELCO^®^, Fresno, CA, USA) and dried under a flow of nitrogen. The images were processed with the open-source Fiji software to calculate the size distribution plots from the Feret diameter.

### 2.3. Synthesis of Doped and Undoped Cdots

The Eu-Cdots were synthesized by the hydrothermal method. Briefly, 0.4 g of citric acid, 0.4 g of urea and 0.3 g of Eu (NO_3_)_3_ were dissolved in 10 mL Milli-Q water. The mixture was transferred into a 50 mL Teflon-lined stainless-steel autoclave where it was heated at 160° for 12 h. The final product was centrifuged at 15,000 rpm during 5 min (three times) to remove the solid contents. The supernatant was diluted in 10 mL of water and dialyzed using a 1 kDa dialysis bag of cellulose acetate (standard grade membrane of regenerated cellulose, Spectra/Por^TM^, Spectrum™ 132638, Thermo Fisher Scientific, Waltham, MA, USA) against deionized water for 8 to 12 days, until no color could be observed in the dialysate. This procedure is used to remove the molecular products present alongside the Cdots and uncomplexed Eu^3+^ in solution. Upon dialysis, the solid Eu-Cdots were collected by lyophilization. For the preparation of control Cdots without Eu^3+^, a similar procedure was followed with no addition of Eu (NO_3_)_3_ to the reaction mixture. On average, the overall yield of the process is only 1.0–2.0 wt% for both procedures.

## 3. Results and Discussion

### 3.1. Structural and Morphological Characterization 

Europium doped Cdots where prepared by a hydrothermal process at 160 °C for 12 h using citric acid, urea and Eu (NO_3_)_3_ as a source of carbon, nitrogen, and europium, respectively. Undoped Cdots were prepared following the same procedure in the absence of europium salt. The TEM images of the Eu-Cdots and the Cdots are shown in Figure 1, together with the Raman and FTIR spectra.

The size distribution of Cdots, with an average diameter of 3.2 ± 0.1 nm (Figure 1c), is typical of the synthesis by the bottom-up procedure. The size distribution of the Eu-Cdots (Figure 1d) is broader with a larger average diameter centered at 5.8 ± 0.5 nm. Similar findings were reported recently for Eu-Cdots with an incorporation of Eu^3+^ of the order of 6% (*w*/*w*) [31]. The increase in size was attributed to the incorporation of the Eu^3+^ ions in the Cdots with a concomitant increase in the disorder induced in the carbon network for high percentages of Eu^3+^, as confirmed by both Raman and X-ray diffraction. The Raman spectra of the Cdots and Eu-Cdots shown in Figure 1e) confirms that there was a slight increase in the disorder of carbon network upon incorporation of Eu^3+^ ions. The Raman spectra is dominated by the characteristic D and G bands of graphitic materials. The D band at 1347 cm^−1^ (Cdots) and 1360 cm^−1^ (Eu-Cdots) is attributed to the sixfold breathing mode of the aromatic rings, and the G band at 1576 cm^−1^ (Cdots) and 1579 cm^−1^ (Eu-Cdots) is attributed to the in-plane bond-stretching motion of pairs of sp^2^ carbon atoms [32,33]. The I_D_/I_G_ integrated intensity ratio was estimated to be 1.9 for the Cdots and 1.3 for the Eu-Cdots. This I_D_/I_G_ of the Eu-Cdots is slightly larger than the values reported for Eu decorated graphene (1.02) [34] and three-dimensional europium-complexed with reduced graphene oxide (1.05) [35]. Within the amorphous carbon regime, characterized by an I_D_/I_G_ increasing with the square of the diameter of the sp^2^ cluster, the Eu-Cdots appear to have a smaller cluster diameter than the Cdots [32]. The average cluster size, estimated using the Ferrari and Robertson relationship for amorphous carbon, [32] is 1.9 nm for the Cdots and 1.6 nm for the Eu-Cdots. The average cluster size for the Cdots is closer to the average diameter of the Cdots estimated by TEM (~3 nm) suggesting that these dots are largely constituted by a pristine core of sp^2^ carbons. In contrast, the Eu-Cdots have an average sp^2^ cluster size that is more than three times smaller than their overall diameter estimated by TEM (5.8 nm). The Raman data suggests a mechanism of formation of the Eu-Cdots that involves a nucleation step induced by coordination of citric acid and urea to Eu^3+^ in solution followed by the typical dehydration and condensation steps that make the sp^2^ network grow around the nucleation center. Thus, we expect that Eu^3+^ is incorporated inside the carbon network in Eu-Cdots, as well as coordinated with suitable functional groups at the edge of the Cdots. 

The FTIR spectra of both Cdots and Eu-Cdots (Figure 1f) show the characteristic broad absorption band centered at 3400 cm^−1^ due to the stretching of the OH (ν_OH_) of the carboxylic and/or hydroxyl groups. Adsorbed water at the surface of Cdots might also contribute to this broad absorption band. What are also common to both materials are the C=C stretching (ν_C=C_) of the sp^2^ clusters in the core at ca. 1578 cm^−1^, and the highly coupled COH bending (δ_COH_) and C-O stretching (ν_C-O_) modes that usually give two characteristic bands with variable relative intensity at ~1400 cm^−1^ and ~1200 cm^−1^ [36]. Most importantly, the relatively sharp and intense band of the C=O stretching (ν_C=O_) of the carbonyl group observed at 1709 cm^−1^ in the Cdots is absent in the Eu-Cdots. This observation suggests that in Eu-Cdots the Eu^3+^ is coordinated via the oxygen atom of the carbonyl and carboxylate groups. The asymmetric and symmetric O-C-O stretching bands of carboxylates coordinated with Eu^3+^ appears at lower wavenumbers, ca 1575 cm^−1^ (νOCO_as_) and 1400 cm^−1^ (νOCO_s_). The former overlaps with the ν_C=C_ mode at 1578 cm^−1^. In addition, several sharp features appear in the 600–900 cm^−1^ spectral region of Eu-Cdots (ca 680 cm^−1^ and 850 cm^−1^) previously associated with Eu-O vibrational modes due to covalent coordination of Eu with oxygen [37].

What is also noteworthy is the absence in the Eu-Cdots of the band observed at 3175 cm^−1^ in the Cdots that we assign to the symmetric N-H stretching (νNH_2_,s) of amides. In addition, in the amide I (1700–1650 cm^−1^) and amide II (1600–1550 cm^−1^) region, Eu-Cdots show a relatively uncomplicated spectra with only one broad band peaking at 1578 cm^−1^ that, as discussed above, is likely to have a contribution from both the νC=C and νO-C-Oas. Conversely, a closer look at the complex envelop of bands observed in the Cdots in this region allows us to identify contributions at 1667 and 1635 cm^−1^ that could be due to the strong amide I and amide II bands. The absence of the amide bands in Eu-Cdots indicate that the incorporation of nitrogen in Eu-Cdots is lower than in the Cdots. This observation is further confirmed by the XPS data shown in Figure 2.

The XPS spectrum of Eu-Cdots (Figure 2b) shows the peaks characteristic of Eu^3+^ at 1135 eV (Eu3d_5/2_), 1165 eV (Eu3d_3/2_), and 137 eV (Eu 4d) in addition to the peaks of C1s, N1s, and O1s around 285, 400, and 532 eV, also present in the Cdots (Figure 2a). The atomic percentages calculated based on the area of the peaks are indicated in Figure 2. The results show that the Eu-Cdots have a higher percentage of oxygen and a lower percentage of nitrogen (39.1% O and 1.2% N) than the Cdots (30.9% O and 7.6% N) that we attribute to the high affinity of Eu^3+^ to the carboxylic groups of citric acid and carbonyl groups of urea favoring the retention of oxygen during the formation of the carbon network.

The high resolution C1s band in Eu-Cdots (Figure 2f) can be deconvoluted in four peaks, with binding energies at: 285.0 eV due to C=C/C-C; 286.6 eV due to hydroxyl (C-OH) and epoxy (C-O-C); 288.6 eV attributed to carboxylate (COO^−^) and the peak at 289.7 eV due to carbon in a more electronegative chemical environment, such as in the carbonate ion (CO_3_^2−^) [38,39]. Many of these carboxylates and carbonates are likely to be coordinated with Eu^3+^. In the C1s region, the major difference between the XPS spectra of doped and undoped dots is the significant decrease of the carbonate band in the Cdots (Figure 2c). The peaks at 286.8 eV and 288.6 eV are relatively unaffected by the coordination with Eu^3+^, because in the Cdots these bands have a smaller contribution from COO^−^ and C-O-C, but they include a contribution from N-C=O and C=O, respectively. Appendix A collects the contribution of each band to the high-resolution spectra in each region. Due to the high number of oxygenated species at the surface of the Cdots and Eu-Cdots, with a strong contribution from carboxylates in both, and carbonates in Eu-Cdots, a negative surface charge can be inferred in both cases.

The O1s spectra of both Cdots and Eu-Cdots can be deconvoluted in three peaks. In Eu-Cdots (Figure 2g), the intense peak at 531.6 eV is attributed to the two equivalent oxygens of carboxylate and carbonate ions, some of which will be coordinated with Eu^3+^. The peak at 532.8 eV is due to the C-O-C of the carboxylate and the C-OH. At higher binding energies, we find a minor contribution from adsorbed water. In the Cdots (Figure 2d), the most intense band is due to the C=O from amides and carboxylic acids at binding energies of 531.5 eV, the COH peaks at 532.6 eV, and there is also a contribution from adsorbed water at higher energies [40].

The N1s spectrum of Cdots can be deconvoluted in three peaks with the major contribution from sp^3^ N at 400.2 eV (amides). Smaller contributions from sp^2^ N at 398.8 eV and quaternary N at 402.0 eV are also observed. The same three contributions, albeit with different relative intensities, can be found in Eu-Cdots, which have a considerably lower amount of nitrogen.

### 3.2. Optical Properties

Figure 3 shows the UV-vis absorption and the luminescence spectra of the Eu-Cdots and compares it with data for isolated Cdots and Eu (NO_3_)_3_ and a mixture of Cdots and Eu (NO_3_)_3_ in water (Cdots + Eu^3+^).

The absorption spectrum of Cdots (Figure 3a) shows a broad and relatively unstructured absorption extending from the UV through the visible region with a characteristic shoulder ~350 nm with contributions from n–π* transition involving the O or N atoms of the functional groups of the Cdots and π–π* transition within the sp^2^ core possibly involving interlayer charge transfer [41]. The UV-vis absorption spectrum of the Cdots + Eu^3+^ mixture is the sum of the spectrum of the isolated Cdots and Eu (NO_3_)_3_ in aqueous solutions. The spectrum of the mixture is dominated by a strong and relatively broad band centered at ~300 nm due to electron transfer from the oxygen to Eu^3+^. This charge transfer band is observed in the Eu (NO_3_)_3_ aqueous solutions whose spectrum is shown in Figure 3a. In addition, the mixture shows a small and sharp peak at 395 nm attributed to the direct absorption of the most intense ^5^L_6_←^7^F_0_ transitions of Eu^3+^ ions that is also observed in the spectrum of Eu (NO_3_)_3_ aqueous solutions [42,43]. The Laporte’s parity selection rule that forbids transition between same parity states is relaxed under the influence of a ligand-field due to non-centrosymmetric interactions that allow the mixing of electronic states of opposite parity into the 4f wavefunctions [43]. Thus, in aqueous solutions the intraconfigurational f–f transitions are partially allowed, and weak absorption and emission can be observed between f-states. Weaker f–f transitions are absent in both the Cdots + Eu^3+^ mixture and the aqueous solution of Eu (NO_3_)_3_. In the mixture, the strong band at 300 nm and the sharp peak at 395 nm due to Eu^3+^ appear superimposed on a broad and continuous absorption extending from the UV through the visible region due to the Cdots. The doped Eu-Cdots have an absorption spectrum that differs significantly from that of the Cdots + Eu^3+^ mixture. The strong absorption at 300 nm that dominated the absorption spectrum of the Cdots + Eu^3+^ mixture is absent in the Eu-Cdots. The absorption spectrum shows a continuous absorption extending from the UV to the red part of the visible spectrum with a shoulder at 350 nm that coincides with that observed in the Cdots.

The emission spectra of Eu-Cdots shows the characteristic sharp peaks at 592, 616 and 698 nm due to the ^5^D_0_→^7^F_J_ (J = 1, 2, 4) transition of Eu^3+^ [43,44] in addition to the broad emission of Cdots centered at 450 nm (Figure 3b). The emission of Eu-Cdots differs from that of the mixture in the relative intensity of the Eu^3+^ sharp peaks above 550 nm that are known for being extremely sensitive to the environment. In Eu-Cdots, the ^5^D_0_→^7^F_2_ transition at 616 nm is the strongest ^5^D_0_→^7^F_J_ transition, whereas in the mixture this peak appears weaker than the peaks observed at 592 and 698 nm due to the ^5^D_0_→^7^F_1_ and ^5^D_0_→^7^F_4_ transitions, respectively. Indeed, the ^5^D_0_→^7^F_2_ transition is considered a hypersensitive transition whose strength is particularly sensitive to the nature of the ligand field. Note that the peak at 720 nm in Eu-Cdots should not be considered in the discussion for it is an artifact from leaking 2nd harmonic of the excitation light at 360 nm.

The origin of the relative intensity changes of the Eu^3+^ centered emission can be elucidated by looking at the excitation spectra collected at 700 nm shown in Figure 3c. This figure shows that direct excitation of the metal-to-ligand charge transfer transition at 300 nm does not result in emission as this band is absent in the excitation spectra of both the Eu (NO_3_)_3_ aqueous solutions and the Cdots + Eu^3+^ mixture. The excitation spectra of the Eu (NO_3_)_3_ aqueous solutions and that of the Cdots + Eu^3+^ mixture are similar showing only the sharp features due to direct excitation of the ^5^L_6_←^7^F_0_ transitions of Eu^3+^. Conversely, in Eu-Cdots, the fact that there is an intensity redistribution in the emission of Eu^3+^ hints to a different excitation mechanism possibly sensitized by the Cdots. The sharp emission bands of the Eu^3+^ in Eu-Cdots result from an indirect excitation by energy transfer (Dexter type exchange mechanism) [45] from the electronic excited sates of the Cdots [31,45,46]. This interpretation agrees with the excitation spectra of Eu-Cdots collected at 700 nm showing a broad structureless excitation consistent with absorption of the Cdots, in addition to the sharp band due to direct excitation of the ^5^L_6_←^7^F_0_ of Eu^3+^. Note that such a broad excitation feature is absent in the spectrum of the Cdots + Eu^3+^ mixture, where no sensitized emission occurs (Figure 3c).

The characteristic broad emission band of Cdots centered around 450 nm is observed in all the three samples containing Cdots (undoped Cdots, Cdots + Eu^3+^ mixture and Eu-Cdots). The emission is excitation wavelength dependent with the maximum of emission shifting towards the red as the excitation energy decreases (see spectrum in Appendix A) for the variation of the emission with the excitation wavelength in Cdots and Eu-Cdots, respectively). This trend is usually associated with heterogeneity of the Cdots either due to the size distribution of the sp^2^ domains or due to the heterogeneous composition of the emitting sites [47]. The luminescence quantum yield of Cdots are quite low (Φ Cdots = 0.016) as expected for emission localized within the sp^2^ carbon core that is free from molecular fluorophores [48]. The slightly lower luminescence quantum yield of Eu-Cdots (Φ_Eu-Cdots_ = 0.013) could be due to energy transfer from the Cdots to the weakly emissive Eu^3+^. The low photoluminescence quantum yield observed are in good agreement with values reported earlier for Eu-decorated graphene quantum dots (Eu-GQD) with Φ_Eu-GQD_ = 0.03 [37]. The photostability of the nanomaterials was evaluated upon prolonged irradiation at 330 nm using the lamp of the spectrofluorometer. In Cdots, a drop of 30% intensity upon 10 min of irradiation was observed due to degradation of some labile defect. Beyond that, no further degradation was observed. For Eu-Cdots, with an intensity drop in the emission of only 15% upon 1 h of irradiation in the same conditions, the degradation kinetics is slower and continuous. Appendix A was introduced in the Appendix A showing the emission intensity upon prolonged irradiation.

### 3.3. Quenching of Emission by Silver and Mercury

Figure 4 shows the effect of addition of an excess of metal cations (100 μM) on the luminescence intensity of Cdots and Eu-Cdots. The plot shows the ratio between the peak intensity at 450 nm before (*I*_0_) and after (*I*) addition of the metal cation upon excitation at 396 nm. Figure 4 shows that the emission of Eu-Cdots is quenched by the presence of Ag^+^ and Hg^2+^. There is a quenching by more than a factor of 25 in the Hg^2+^ and a more modest quenching effect (factor of 2) is induced by Ag^+^. Except for these two cations, the Eu-Cdots are relatively insensitive to all the other cations tested. Similar results were obtained for excitation in the 280–400 nm range. The Cdots dispersion produced in the absence of Eu^3+^ is insensitive to all the cations including Ag^+^ and Hg^2+^ (Figure 4).

The enhanced sensitivity to the presence of metal cations of Eu-Cdots, with respect to undoped Cdots, reflects the differences in structure and availability of active sites for cation binding of the two nanomaterials. The TEM images show that the average size of the Eu-Cdots is larger than that of undoped Cdots suggesting that there is an incorporation of Eu^3+^ ions in the graphene sheets during the growth of the sp^2^ network. It is possible that the Eu^3+^ becomes part of the carbon layer due to coordination with the carboxylic groups of the citric acid and the carbonyl groups of urea during the condensation reactions to form the sp^2^ clusters [49]. Eu^3+^ acts as a nucleating center expanding the overall size of the Cdots. In this case it is not unreasonable to expect a smaller size for the sp^2^ clusters in each layer due the presence of oxygenated defects induced by incorporation of Eu^3+^ in the carbon network, as it was effectively estimated based on the Raman spectroscopy data. Furthermore, the incorporation of Eu^3+^ in the Cdots might also affect the structure of the Cdots by inducing interaction between stacks of layers with different orientation. Recent computational studies have shown that the structure of the Cdots is influenced by the number of layers of sp^2^ carbons, with a uniaxial stacked structure observed for a low number of layers (up to 5), while for larger numbers of layers, the stacking breaks down into multi-axial stacked structures, where stacked layers with different orientations coexist. [50] The Eu^3+^ ion can operate as a seed for multiaxial stacking, as it was predicted earlier for K^+^ ions and coronene clusters [50]. Furthermore, due to its small ionic radius (~0.95–1.12 Å depending on the coordination), [51] Eu^3+^ can also diffuse to some extent and intercalate between layers typically separated by a distance of 3.3 Å. Thus, there are several mechanisms by which the Eu-Cdots could incorporate Eu^3+^ ions in its structure affecting its response to the presence of metal cations in solution.

The Ag^+^ and Hg^2+^ quenching effect on the emission spectra of the dispersion of Eu-Cdots is shown in detail in Figure 5. For Hg^2+^, both the broad emission of the Cdots at 450 nm and the sharp emission bands of Eu^3+^ ions decrease in intensity with the amount of Hg^2+^ in solution, while the spectral shape is maintained (Figure 5a). In contrast, for the Ag^+^ ion (Figure 5b), quenching is observed in the emission of the Cdots, while the sharp Eu^3+^ emission peaks remain almost constant. This specificity in the quenching mechanism allows us to differentiate the two ions in solution. Furthermore, the constant intensity of Eu^3+^ for an increasing concentration of Ag^+^ can be used as an internal reference to build a ratiometer optical sensor for Ag^+^. Ratiometer sensors are less prone to artifacts and can correct for experimental and environmental effects, which are particularly problematic for ion sensing in complex biological matrixes where the concentration of the sensor can vary locally [52].

The plot of the intensity ratio (*I*_0_/*I* − 1) of the Eu-Cdots emission (λ_em_ = 450 nm) as a function of the concentration of Hg^2+^ is linear up to the concentration of 80 µM (Figure 5c), obeying a Stern–Volmer type equation:(1)$$\frac{I_{0}}{I}-1=K_{SV}\left[{\mathrm{Hg}}^{2+}\right]$$
where $K_{SV}$ is a static quenching constant of $K_{SV}$ = (2.8 ± 0.1) × 10^5^ M^−1^. The limit of detection, estimated from the standard deviation of the straight-line intercept and the slope (LOD = 3σ/*K_SV_*), is 4 µM. The Stern–Volmer constant for the quenching of the Eu ^3+^ emission (λ_em_ = 698 nm) is one order of magnitude lower, $K_{SV}$ = (1.7 ± 0.1) × 10^4^ s^−1^, but still one order of magnitude larger than $K_{SV}$ for the dispersion of undoped Cdots (3.6 ± 0.5) × 10^3^ s^−1^.

For the Ag^+^ ion, the Stern–Volmer plot of the variation of the fluorescence intensity ratio (*I*_0_/*I* − 1) at 450 nm with the concentration of Ag^+^ in water shows a downward curvature (Figure 5d). This trend indicates that not all quenching centers are readily accessible for the Ag^+^ ions. In this case, considering that the fraction of active quencher centers is *f_a_*, a linear relationship is expected for the variation of *I*_0_/(*I*_0_ − *I*) with the reciprocal of the Ag^+^ following Equation (2): [53]
(2)$$\frac{I_{0}}{I_{0}-I}=\frac{1}{f_{a}K_{SV}}\times \frac{1}{\left[{\mathrm{Ag}}^{+}\right]}+\frac{1}{f_{a}}$$

The inset in Figure 5d shows that such a linear trend is indeed observed for the quenching of the Eu-Cdots emission by Ag^+^. The fitting of the experimental data to Equation (2) gives for the fraction of active sites *f_a_* = 0.60 ± 0. 2 and *K*_*SV*_ = (2.4 ± 0.4) × 10^5^ s^−1^. Similar values for *K_SV_* were obtained for excitation at different wavelengths (e.g., λ_exc_ = 280 nm, *f_a_* = 0.40, *K_SV_* = 5.0 × 10^5^ s^−1^ λ_exc_ = 360 nm, *f_a_* = 0.47, *K_SV_* = 1.5 × 10^6^ s^−1^). The *K_SV_* value for the mixture of Cdots + Eu^3+^ is 3.3 × 10^4^ s^−1^ (*f_a_* = 0.58) and for undoped Cdots is lower than 10^3^ s^−1^. The LOD was estimated as ~5 µM for Ag^+^.

### 3.4. Mechanistic Insight

The Hg^2+^ cation can accommodate a range of coordination numbers and geometries being the two-coordinated linear and four-coordinated tetrahedral very common [23]. Hg^2+^ is a soft acid that likes to coordinate with soft bases containing sulfur, phosphorous, and halides [30]. In the case of Cdots, the Hg^2+^ coordinates with free oxygen and nitrogen atoms at the surface of Eu-Cdots, and it can also replace Eu^3+^ ions coordinated at the surface. In addition, since the radius of Hg^2+^ (~0.60–1.14 Å depending on the coordination) [51] is similar to that of Eu^3+^ radius (~0.95–1.12 Å), it can also access the interlayer space and replace the Eu^3+^. Interaction of Hg^2+^ with Cdots can cause static quenching of the Eu-Cdots emission by several mechanisms, including the heavy atom spin–orbit coupling enhancement and charge transfer mechanisms [54]. The concomitant quenching of Eu^3+^ confirms that some of the Eu^3+^ ions are indeed displaced by Hg^2+^, in which case the sensitization of Eu^3+^ emission by energy transfer from the Cdots is no longer possible.

The Ag^+^ cation is a soft acid like Hg^2+^ with a radius of~0.67–1.28 Å that is slightly larger than that of Hg^2+^ and Eu^3+^ for the same coordination numbers [51]. Similarly to Hg^2+^, Ag^+^ also prefers to coordinate with soft bases, it favors linear or tetrahedral coordination, and it should also be able to access the interlayer space. The interaction of the Ag^+^ ions with the Cdots must involve coordination of the Ag^+^ with the carboxylates of the Cdots, and possibly also the minor amount of nitrogen containing groups. The fact that the intensity of Eu^3+^ remains almost constant for all Ag^+^ concentrations indicates that the Ag^+^ ions are unable to displace the Eu^3+^ coordinated with the Cdots. Thus, interactions of Ag^+^ with the Cdots must be mostly via free carboxylates. In this scenario, the number of active quenching sites, given by the fraction *f_a_* = 0.4–0.6, must be related with the amount of free carboxylate at the periphery of the Eu-Cdots. The quenching of the emission of Cdots upon complexation can occur by both the heavy atom spin-orbit coupling enhancement and the electron or charge transfer from Ag^+^ to the Cdots active sites. Most importantly, since the emission of Eu^3+^ remains unchanged, the ratio of the Cdots to the Eu^3+^ emission intensities can be used as an internal reference to build a ratiometer sensor for Ag^+^ that is less prone to artifacts and can correct for environmental effects.

Additional insight into the sensing mechanism can be retrieved from the emission decay curves shown in Figure 6 with the intensity in a log scale to better convey the details of the curves. The same curves are shown in Appendix A, in a normal scale for longer times. The decay curves of the Eu-Cdots emission in the absence and in the presence of Ag^+^ and Hg^2+^ cations are complex and can only be fitted by a sum of 3 exponentials plus a very short lifetime component (~10 ps) attributed to light scattering. The average lifetime of the Cdots (3.2 ns) is slightly higher than that of the Eu-Cdots (2.95 ns) reflecting the energy transfer from the Cdots to the Eu^3+^ ions. The average lifetime of Eu-Cdots is shorter in the presence of 100 µM of Ag^+^ (2.2 ns). This indicates that, in addition to static quenching by formation of non-emissive complexes, the quenching by Ag^+^ has a dynamic component due to the weak coordination of the Ag^+^ to the oxygen of the Eu-Cdots functional groups allowing for some Ag^+^ ions to remain free in solution. The opposite trend is observed in the presence of Hg^2+^ where the lifetime increases to 7.8 ns. This is a rather unexpected observation given the strong quenching of the emission of Eu-Cdots in the presence of Hg^2+^. The main interaction of the Hg^2+^ with Eu-Cdots is the one leading to static quenching due to formation of non-emissive complexes via the carbonate and carboxylate groups. The stronger quenching in the presence of Hg^2+^ is correlated with the possibility of Hg^2+^ to replace Eu^3+^ not only at the surface of the dots but also in the interlayer spacing, which must require a fine balance between charge, size of the ions and also the preferred coordination geometry. Thus, the increase in the excited state lifetime can only be explained by a decrease of the nonradiative deactivation channels of uncomplexed Eu-Cdots upon transient interaction with the Hg^2+^ that remains in solution. The hydroxyl groups of the Cdots are strongly coupled with high frequency vibration modes of water via hydrogen bonding contributing to dissipation of the electronic excitation energy into the bath states of the solvent. The Hg^2+^ that remains in solution can transiently interact with the hydroxyl groups inhibiting this nonradiative deactivation channel and increasing the Eu-Cdots average lifetime.

## 4. Conclusions

Carbon dots and Eu-Cdots were prepared by a hydrothermal treatment using citric acid and urea as precursors and Eu (NO_3_)_3_ as a source of europium. The Eu^3+^ cation is coordinated with the oxygen of the functional groups at the Cdots surface and incorporate into the carbon network, forming Eu-O charge transfer complexes. It can also be intercalated between the nanographene sheets. The Eu-Cdots can be used as turn-off luminescent sensors for Hg^2+^ and Ag^+^ cations because their luminescence is strongly quenched in aqueous solutions in 10–100 µM range concentrations with a limit of detection of 4–5 µM

For Hg^2+^, both the broad emission of the Cdots at 450 nm and the sharp emission bands of Eu^3+^ ions are quenched, while for the Ag^+^ ion, quenching is observed only in the emission of the Cdots. The specificity in the quenching mechanism allows us to differentiate the two ions in solution. The constant intensity of Eu^3+^ for an increasing concentration of Ag^+^ can be used as an internal reference to build a ratiometer optical sensor for Ag^+^. It is worth noting that this ratiometric sensor can be of value in clinical mercury poisoning detection in complex biological matrixes, but further work should be done to improve the detection limits. Except for Hg^2+^ and Ag^+^, the Eu-Cdots are relatively insensitive to all other cations tested. The Cdots dispersion produced in the absence of Eu^3+^ is relatively insensitive to all cations, including Ag^+^ and Hg^2+^. This increase of reactivity of the Eu-Cdots when compared with undoped Cdots is associated with the differences in the composition of the doped dots (more oxygenated, with additional carboxylate groups) and availability of active sites for cation binding.

## Figures and Tables

**Figure 1 nanomaterials-12-00385-f001:**
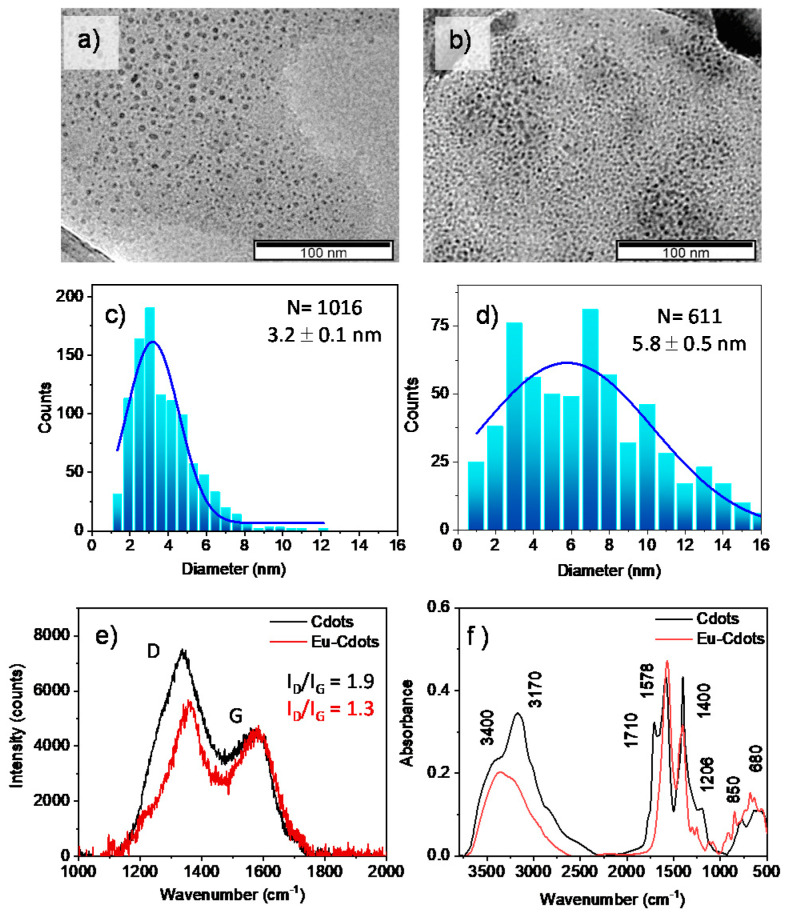
Characterization of Cdots and Eu-Cdots by TEM (**a**–**d**), Raman (**e**), and FTIR (**f**); (**a**,**b**) TEM images of Cdots and Eu-Cdots, and (**c**,**d**) their corresponding distributions of Ferret diameter; (**e**) Raman spectra of the Cdots (black) and Eu-Cdots (red) showing the characteristic D and G bands, and (**f**) FTIR spectrum of Cdots (black) and Eu-Cdots (red).

**Figure 2 nanomaterials-12-00385-f002:**
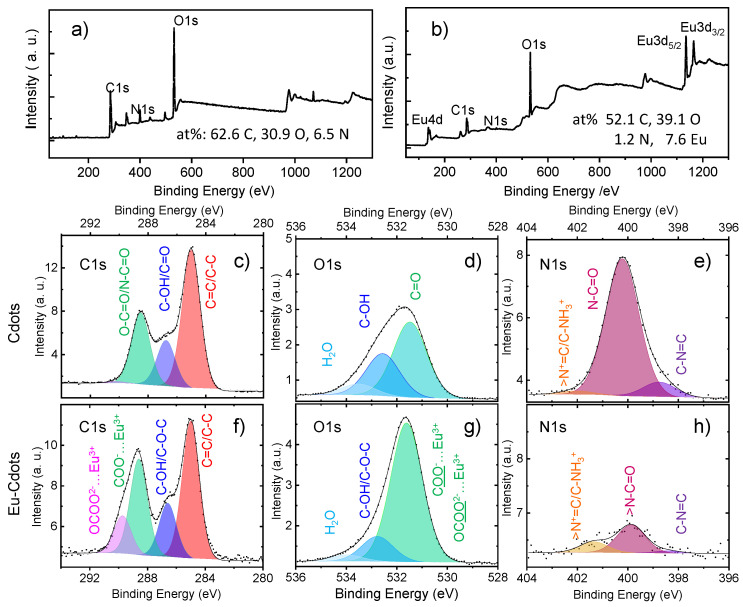
XPS spectra of the Cdots and Eu-Cdots: low resolution XPS spectra (**a**,**b**) and high resolution XPS spectra for the C1s, O1s, and N1s peaks of the Cdots (**c**–**e**) and Eu-Cdots (**f**–**h**).

**Figure 3 nanomaterials-12-00385-f003:**
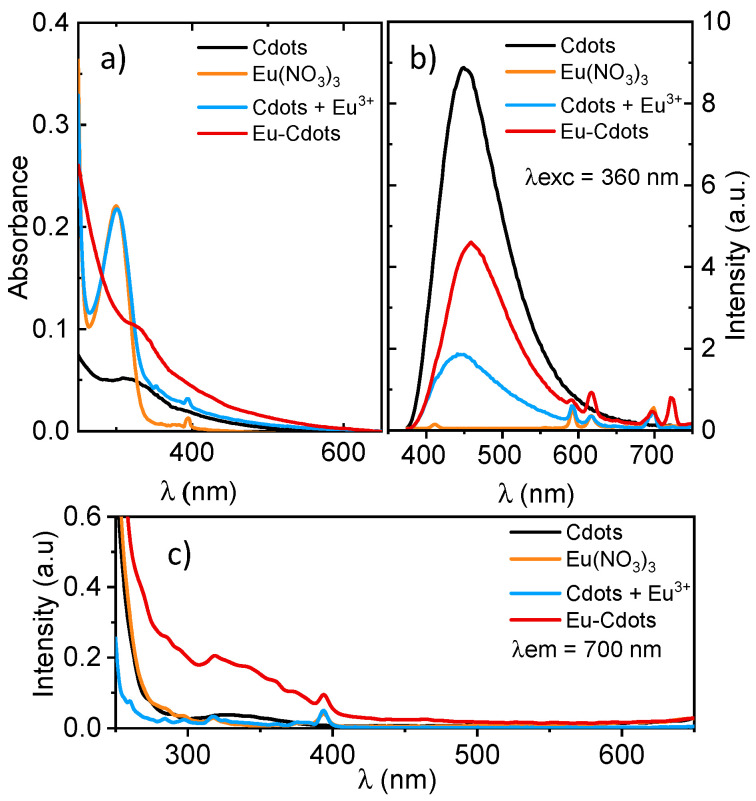
Optical properties of Cdots (black), Eu (NO_3_)_3_ (orange), a mixture of Cdots + Eu^3+^ in water (blue) and Eu-Cdots (dark red); (**a**) absorption spectra, (**b**) photoluminescence emission spectra excited at 360 nm, and (**c**) photoluminescence excitation spectra collected at 700 nm.

**Figure 4 nanomaterials-12-00385-f004:**
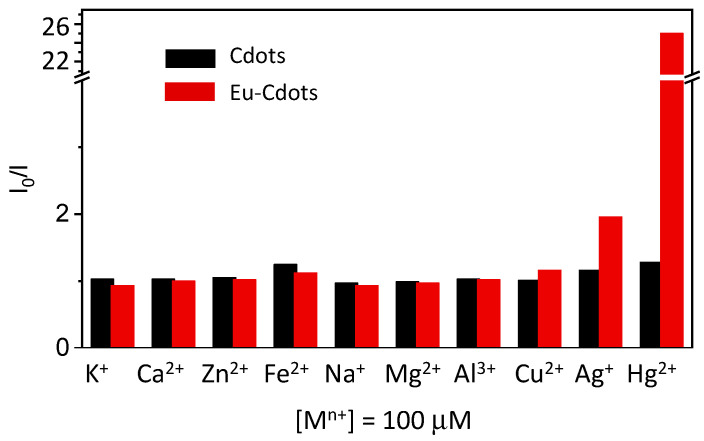
Ratio of the emission intensities before (*I*_0_) and after (*I*) addition of 100 μM of several metal cations upon excitation at 396 nm of a Cdots (**black**) and Eu-Cdots (**dark red**). A break was introduced in the *y*-axis to better convey the details of the response. A quenching of emission by a factor of 2 is observed for Ag^+^, while Hg^2+^ shows a quenching by a factor of 25.

**Figure 5 nanomaterials-12-00385-f005:**
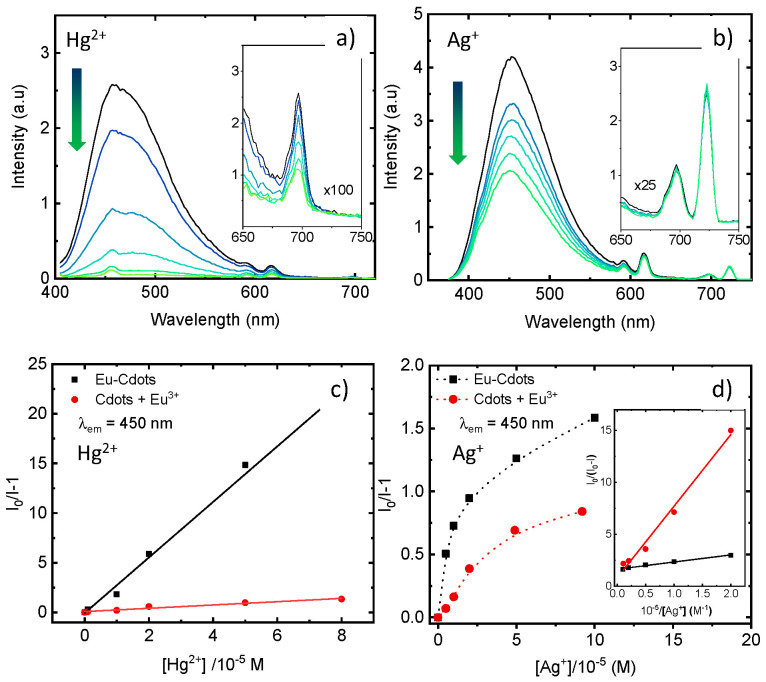
Emission quenching of Eu-Cdots by Hg^2+^ and Ag^+^. Emission spectra of the Eu-Cdots aqueous dispersion in the presence of Hg^2+^ at λ_exc_ = 390 nm (**a**) and Ag^+^ at λ_exc_ = 360 nm (**b**), and corresponding Stern–Volmer plots of the quenching measured at λ_em_ = 450 nm upon addition of Hg^2+^ (**c**) and Ag^+^ (**d**). In (**c**,**d**), similar quenching data are shown for the quenching of Cdots + Eu^3+^ mixture. The inset on panel (**d**) shows linear relationship between *I*_0_/(*I*_0_ − *I*) with the reciprocal of the Ag^+^ following Equation (2).

**Figure 6 nanomaterials-12-00385-f006:**
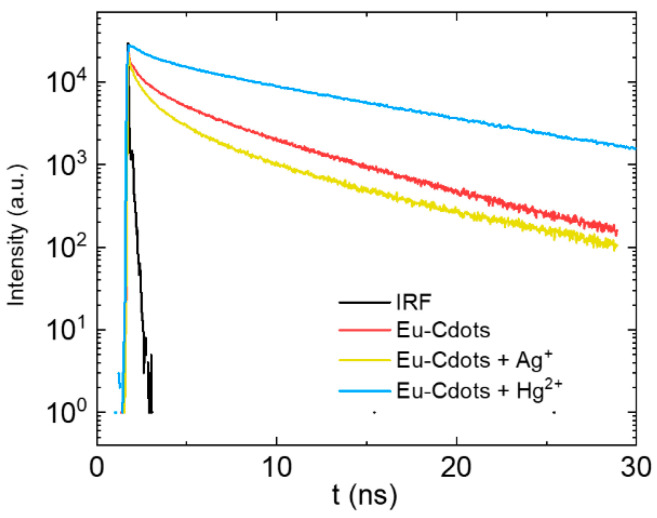
Emission decay curves for Eu-Cdots (**red**), and Eu-Cdots in the presence of 100 mM of Ag^+^ (**yellow**) and Hg^2+^ (**blue**). The instrumental response function (IRF) is shown in black.

## Data Availability

The data presented in this study are available upon reasonable request from the corresponding author.

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
