# Peer review of "A Fluorescent Nanosensor for Silver (Ag+) and Mercury (Hg2+) Ions Using Eu (III)-Doped Carbon Dots"

_nanomaterials, 2022, doi:10.3390/nano12030385_

Round 1

Reviewer 1 Report

1.the author said for the Ag+ ion (Fig 6b), quenching is observed in the emission of the Cdots, while the sharp Eu3+ 368 emission peaks remain almost constant, but in the Fig.5, it is not the same. why? 2.The real application should be supplied.

Author Response

We thank the referee for taking the time to review our work. The referee’s comments and suggestion contributed significantly to improve our revised manuscript.

  1. Thanks to the referee’s comments we realized that the caption on figure 5 was not correct. We have corrected the caption in the revised manuscript. Figure 5, like Fig 6b, shows a quenching of emission in the presence of silver. The reason why the quenching is not so evident in Fig 5 as in Fig 6 is the scale. In fig 5 the much stronger quenching induced by Hg2+ sets the upper limit of the y-axis too high. To better convey the results, we added a break in the y-axis in the revised Fig. 5. In the revised figure the more modest quenching in the presence of Ag+ is now evident.
  2. Given the relatively high limit of detection estimated for both cations sensing using Eu-Cdots is not yet a competitive strategy that is worth exploring in a real application. The relevance of the work relys on the fact that this strategy allows for ratiometric sensing that can be of value in clinical mercury poisoning detection in complex biological matrixes. However, further work should be done to improve the detection limits, which we believe relates to the amount of Eu3+ within the carbon network. We have clarified this aspect in the conclusion.

Reviewer 2 Report

This study on the use of carbon dots and Eu-functionalized carbon dots for detection of Ag+ and Hg+ ions via luminescence quenching is well done and of large interest in the field of nanomaterial applications.  I really only have a few minor questions about the work itself:

1) Beyond 20 ns, do the emission decay curves return to the baseline value?  Or are there long term or permanent changes?

2) Were experiments performed that tested cations of Ag and Hg as the same time?  I would be interested to see the combined response and the ratiometer calculations in practice to gauge the desired applications.

Further, are there any problems such as degradation or decomposition under 360 nm excitation for extended time periods (Cdots, Ag, or Hg)?

Author Response

We thank the referee for taking the time to review our work. We appreciate the comments and suggestion that have contributed to significantly improve our revised manuscript. In the following a point-by-point response to the comments of the referee is given

1) Beyond 20 ns, do the emission decay curves return to the baseline value?  Or are there long term or permanent changes?

All the emission curves decay to the noise level. To better convey this trend, in the supporting information we added Figure S4 showing the decays in a normal scale and in a wider timescale. Please note that in the manuscript the decays are presented in a log scale to better convey the details of the curve. The number of counts for Eu-CDots and Eu-Cdots + Ag+ are already quite close to the noise level at 30 ns (100 counts).

2) Were experiments performed that tested cations of Ag and Hg as the same time?  I would be interested to see the combined response and the ratiometer calculations in practice to gauge the desired applications.

We did not perform such experiments because, due to the relatively high limit of detection estimated for both cations, sensing using Eu-Cdots is not yet a competitive strategy that is worth exploring in a real application. However, the work thus provides indications to improve the performance of the sensor, which appears to relate to the amount of Eu3+ incorporated in the carbon network. We have clarified this aspect in the conclusion.

Further, are there any problems such as degradation or decomposition under 360 nm excitation for extended time periods (Cdots, Ag, or Hg)?

We checked the photostability of Cdots and Eu-Cdots upon irradiation at 330 nm using the 450 W fluorimeter lamp. In Cdots, we observed a drop of 30% intensity upon 10 min of irradiation. Beyond that the dots are quite stable. This trend is understood as an initial degradation of some labile defect. For Eu-Cdots, with an intensity drop of only 15% upon 1h of irradiation in the same conditions, the degradation kinetics is slower and continuous. Figure S3 was introduced in the supplementary material showing the emission intensity upon prolonged irradiation. This discussion was introduced on page 8, L330-337

Reviewer 3 Report

The manuscript entitled, ‘A fluorescent nanosensor for silver (Ag+) and mercury (Hg2+) ions using Eu(III) doped carbon dots’ discussed the activity of NCDs for sensing applications. The work is quite interesting and summarized the leading areas. I am mentioning some loopholes of this work which should be accounted prior to publication;

  1. First of all, the aim of the work should be rewritten.
  2. How Eu affect the quantum yield of the CDs? Any reason for this?
  3. Why only Hg is sensed? What is the mechanism behind this? Is it chelation based or some other effects? Those should be cleared.
  4. Some articles based on carbon dots could improve its literature review like, https://doi.org/10.1021/acsanm.0c02305; DOI: 1039/D1NA00447F; https://doi.org/10.1016/j.snb.2017.06.068.
  5. Author wrote ‘…while for larger numbers of layers the stacking breaks down into multiaxial stacked structures.’ What does it signify?
  6. Could the author infer about the surface charge of the CDs prepared for both doped and undoped?

Author Response

We thank the referee for taking the time to review our work. We appreciate the comments and suggestion of the referees that have contributed to clarify some aspects of the report and improve the overall quality of the revised manuscript. In the following a point-by-point response to the comments of the reviewers is given.

  1. First of all, the aim of the work should be rewritten.

    On page 2 L 85-86, we have revised the aim of the manuscript to better convey the scope and novelty of the work.

    2. How Eu affect the quantum yield of the CDs? Any reason for this?

    In page 8, lines 326-330, we discuss that the luminescence quantum yield of Eu-Cdots (ΦEu-Cdots = 0.013)  is slightly lower than that of undoped Cdots due to energy transfer from the Cdots to the weakly emissive Eu3+.

    3. Why only Hg is sensed? What is the mechanism behind this? Is it chelation based or some other effects? Those should be cleared.

    The Eu-Cdots have a much stronger response towards Hg2+, but they can sense the presence of both Hg2+ and Ag+ in different ways. In both cases the emission of the Cdots is quenched, while the emission of Eu3+ is quenched in the presence of Hg2+ and remains unchanged in the presence of Ag+. The mechanism of sensing is discussed in page 11, section 3.4 Mechanistic insight.  Interaction of the dots with the metals involves mainly the formation of non-emissive complexes. The exact mechanism of quenching (spin-orbit coupling and/or charge/electron transfer) cannot be elucidated without further information from transient absorption experiments that is currently beyond our reach. Additionally, the quenching by Ag+ silver has also a contribution from dynamic quenching. The stronger quenching due to Hg2+ quenching is correlated with the possibility of Hg2+ to replace Eu3+ not only at the surface of the dots but also in the interlayer spacing, which must require a fine balance between charge, size of the ions and also the preferred coordination geometry.

    4. Some articles based on carbon dots could improve its literature review like, https://doi.org/10.1021/acsanm.0c02305; DOI: 1039/D1NA00447F; https://doi.org/10.1016/j.snb.2017.06.068.

    Application in sensing of Cdots is indeed a hot topic of research where interesting reviews have been published recently. We have revised the cited references by adding the suggested references.

    5. Author wrote ‘…while for larger numbers of layers the stacking breaks down into multiaxial stacked structures.’ What does it signify?

    “multiaxial stacked structures” means that we start to see multiple stacks with different orientations. The meaning of this expression has been clarified on page 9, line 368

    6. Could the author infer about the surface charge of the CDs prepared for both doped and undoped?

    Due to the high number of oxygenated species at the surface of the Cdots and Eu-Cdots, with a strong contribution from carboxylates in both, and carbonates  in Eu-Cdots a negative surface charge can be inferred in both cases. The relatively low amount of Eu3+(~6 wt%) does not reduce significantly the Cdots surface charge as proven by the stability of the Eu-Cdots dispersion. We have included a comment on the surface charge in page 6, Lines 241-244.

Round 2

Reviewer 1 Report

It can be accepted.

Reviewer 3 Report

The comments are taken care by the authors. Hence, it can be published in its present form.